# Fast and Efficient Evaluation of the Mass Composition of Shredded Electrodes from Lithium-Ion Batteries Using 2D Imaging

**DOI:** 10.3390/jimaging9070135

**Published:** 2023-07-05

**Authors:** Peter Bischoff, Alexandra Kaas, Christiane Schuster, Thomas Härtling, Urs Peuker

**Affiliations:** 1Fraunhofer Institute for Ceramic Technologies and Systems IKTS, Maria-Reiche-Str. 2, 01109 Dresden, Germany; christiane.schuster@ikts.fraunhofer.de (C.S.); thomas.haertling@ikts.fraunhofer.de (T.H.); 2Institute of Solid State Electronics, Technische Universität Dresden, 01069 Dresden, Germany; 3Institute of Mechanical Process Engineering and Mineral Processing, TU Bergakademie Freiberg, 09599 Freiberg, Germany; alexandra.kaas@mvtat.tu-freiberg.de (A.K.); urs.peuker@mvtat.tu-freiberg.de (U.P.); 4Fraunhofer Portugal Center for Smart Agriculture and Water Management—AWAM, Rua Alfredo Allen 455/461, 4200-135 Porto, Portugal

**Keywords:** image segmentation, recycling, separation process, mass composition

## Abstract

With the increasing number of electrical devices, especially electric vehicles, the need for efficient recycling processes of electric components is on the rise. Mechanical recycling of lithium-ion batteries includes the comminution of the electrodes and sorting the particle mixtures to achieve the highest possible purities of the individual material components (e.g., copper and aluminum). An important part of recycling is the quantitative determination of the yield and recovery rate, which is required to adapt the processes to different feed materials. Since this is usually done by sorting individual particles manually before determining the mass of each material, we developed a novel method for automating this evaluation process. The method is based on detecting the different material particles in images based on simple thresholding techniques and analyzing the correlation of the area of each material in the field of view to the mass in the previously prepared samples. This can then be applied to further samples to determine their mass composition. Using this automated method, the process is accelerated, the accuracy is improved compared to a human operator, and the cost of the evaluation process is reduced.

## 1. Introduction

The production of lithium-ion batteries (LIB) has increased during the last years and is continuing to do so. One reason for this is the rising number of electric vehicles (EV), which is predicted to go from 11 Mio in 2020 to approximately 145–230 Mio in 2030 [1]. Therefore, the return flow of LIBs is estimated to be around 1.5 Mio t per year by 2040 [2]. This increasing number is related to both the rising number of EVs and the estimated lifespan of battery packs, i.e., 4.5–14.5 years depending on the operation conditions [3,4]. One challenge with LIBs is the availability of critical and geostrategically important materials such as cobalt, nickel, lithium, and graphite [5,6]. This highlights the importance of recycling LIBs and ensuring a closed material loop to reduce the further exploitation of resources.

Recycling battery cells requires the recycled materials to be high purity to achieve a constantly high battery performance. The graphite used in LIB has a purity of up to 99.9% [7]. Impurities in the recycling products cause decreased ion transport in battery cells produced from the recycled material. Therefore, it is necessary to monitor and control the separation processes.

Introducing sensors into separation processes enables the outcome to be controlled, which optimizes the separation result in situ [8]. Different sensor systems for mechanical recycling were reviewed by Kroell et al. [9]. Optical sensors show a big advantage due to the possibility of large scale applications, low investment and operation costs, and low health risks compared to X-ray [8,10]. Kroell et al. concluded that RGB and hyperspectral sensors are most suited for material flow characterization. They are chiefly applied to differentiate materials and are mostly used for glass, paper, and metal detection in the recycling industry [9,11].

In battery recycling processes, RGB sensors are used to identify the components of disassembled cells: the separator, cathode, and anode [12]. The crushed battery cell products, especially the coarse fraction, need further separation in the housing, separator, and delaminated current collector foils. A combination of metal sensors and RGB cameras is used to purify the separation products after air-flow separation [13,14] using a blow-out system. Moreover, a system of only RGB sensors is used to detect copper and aluminum from the housing fraction and subsequently further separate those components [15,16].

Determining the mass composition before and after separation is critical for a quantitative description of separation products. Currently, the characterization of the separation products of battery recycling is performed by manual sorting [14,15,17,18,19,20,21,22]. This procedure is time consuming and labor and cost intensive [16,23]. Being able to measure the mass composition of samples of mixed materials inline would significantly improve the efficiency and facilitate rational decision-making during the separation process.

Even though this is a problem that could, for example, be solved using modern machine learning and, especially, deep learning technology, those approaches do not only require expert knowledge but are also time consuming to set up and can still be prone to errors. In combination with the hardware required for the training and inference of such models, this can result in cost-intensive projects.

Therefore, this paper aims to calculate overall the mass composition (mass of aluminum particles vs. mass of copper particles) of mechanically processed LIB material from continuously collectible image data using simple computer vision techniques to detect individual particles. This contributes to maximizing the efficiency and lowering the cost of analyzing and validating separation processes. The introduction of the developed system as an inline measurement tool allows for the quantification of the respective separation process.

## 2. Materials and Methods

### 2.1. Material Preprocessing

The battery material in our work originated from automotive cells. Our experiments were performed on the material of a prismatic cell (Samsung SDI, 2019), which was subjected to multiple charging and discharging cycles. More details on the original cell are described by Werner et al. [24].

For the first comminution a low-speed axial-gab rotary shear (0.4 m/s) (developed and built by TU Freiberg 1994) was used to open up the battery [17,25,26]. Two types of comminution tools were used: a V-shaped shear and a typical rotor shear. Subsequently, a one-shaft rotary shear, hereinafter, a granulator (MeWa Andritz Universal Granulator UG 300; Hechingen, Germany) with a 20 mm grid was used to determine the particle size in the first process stage. More detailed information about the machines in the first process stage can be found in [16,17,25,27].

The shredded cell was then dried for 14 days at 24 °C. To remove the released black mass, the material was screened with a sieve size of 1 mm. In order to remove the separator foil and the cell housing, the material was sorted in a 120° angle air classifier with a velocity range of 0–19.04 m/s and nine stages. The separator foil was removed at 2.0 m/s and the cell housing at 4.7 m/s.

A high-speed impact mill (Turborotor Görgens G-35S, Dormagen, Germany) was applied for the second comminution. After the second screening at 0.5 mm, the remaining black mass was removed. Particles larger than 0.5 mm formed the material used for this paper. Figure 1 shows a mixture of those aluminum and copper particles.

### 2.2. Experimental Setup

To achieve a cost-efficient setup compared to manual sorting, we used a single-board computer (Raspberry Pi 4B) in combination with the appropriate camera module (V2) with a combined cost of less than EUR 150. The camera was mounted facing downwards onto the material, covering a field of view (FOV) of approximately 50 mm by 60 mm as shown in Figure 2. The focal length of the camera was 3.04 mm and the focal ratio was 2.0. The built-in sensor (Sony IMX219) had an area of 3.68 mm by 2.76 mm with a physical pixel size of 1.12 μm by 1.12 μm (3280 by 2464 pixels). The working distance $L_{W}$ (the distance between the camera lens and the background on which the particles were placed) was 88.5 mm. The resulting feature resolution $R_{F}$ (the smallest feature, which was reliably resolved by spanning over at least four pixels) was 0.08 mm.

The material samples being analyzed were placed on a bright paper background with the illumination set to avoid large shadows and dark areas as much as possible. We did not use a specialized, external light source but only ceiling lighting and asserted that neither reflections from the surrounding walls nor through sunlight would affect the image quality. The captured images were intermediately saved in the PNG format with a pixel bit depth of 24 bits. The processing was performed in a Python script to analyze the material mass. The analysis is described in more detail in the following sections.

Additional approaches we considered for the material specific segmentation of mixed samples were transfer-learning a pretrained deep learning model designed for instance-segmentation problems (mask R-CNN [28,29]) and using a conventional clustering algorithm (K-means clustering) to assign the pixels to the different material classes after removing the background. Even though we had a relatively large dataset with approximately 500 images with multiple particles each, the deep learning approach failed to accurately segment and classify our particles. We suspect that this was due to the high number of particles (>100) per image in contrast to a relatively small number of objects of each class in the original dataset (“Common objects in Context”, COCO). Therefore, we do not consider deep learning approaches further in this paper.

### 2.3. Calibration from Pure Material Samples

To calibrate our readings and enable the material mass analysis of mixed-material samples (in our case aluminum and copper particles), we first analyzed samples of pure materials, either aluminum or copper. We manually sorted the preprocessed material to achieve pure samples. Both materials were divided into small batches of particles with a cumulative weight less than one gram. Each batch was weighed individually on a precision balance before being placed underneath the camera setup and roughly distributed to avoid the overlap of multiple particles.

The sample particles were segmented from the background using different color spaces depending on the material under consideration. We found that aluminum is best segmentable in images using the YUV color model, which displays colors based on their luminosity (*Y*) and their chrominance (*U* and *V*). The YUV channels can be retrieved from the RGB channels using the following conversion, which we adopted from ImageJ [30]:(1)$$\left(\begin{array}{c} y\\ u\\ v \end{array}\right)=\left[\begin{array}{cccc} 0.299 & 0.587 & 0.114 & \\ -0.169 & -0.332 & 0.5 & \\ 0.5 & -0.419 & -0.0813 & \end{array}\right]\;\cdot \;\left(\begin{array}{c} r\\ g\\ b \end{array}\right)+\left(\begin{array}{c} 0\\ 128\\ 128 \end{array}\right)\;.$$

We manually identified thresholds for the YUV channels using ImageJ. The threshold was set to the following condition:(2)3<Y<187∧126<U<136∧119<V<133.

The resulting binary mask was further processed using morphological closing with a circular neighborhood of six pixels followed by a morphological opening with a circular neighborhood of twelve pixels. Finally, the correlation between the foreground pixels in the binary mask and the mass of the material samples was analyzed using a linear regression model.

The copper particles, on the other hand, were more easily distinguished from the background using the HSV color space (hue, saturation, value). The conversion from an RGB to HSV color space was implemented in *scikit-image* [31]. We also set the thresholds through manual experimentation in ImageJ and found the following expression to work well:(3)(H<30∨H>234)∧S<46∧V<255.

We did not follow this thresholding by morphological operations in the case of the copper particles. Similar to the regression of the aluminum particles, we also used a linear regression model here to determine the correlation between the foreground area and the mass of the copper particles.

### 2.4. Validation Using Mixed-Material Samples with Known Mass Distribution

Before applying this method in practical environments, the correlation between the segmented areas and the mass of the respective particles had to be validated using samples with mixed materials. We, therefore, prepared additional samples by firstly sorting the materials and preparing pure samples and then documenting their mass. The samples were then interspersed and finally placed on the paper background with the exact same camera setup as described above. The images were taken under the same illumination and segmented twice: once in the YUV color space to find the mass of the aluminum particles and once in the HSV color space to find the mass of the copper particles.

The result of the copper segmentation from the background can be seen in Figure 3 in a mixed-material sample. The yellow area in the segmentation mask in Figure 3b marks the copper particles in the input image in Figure 3a.

Subsequent to the segmentation and determination of the foreground area for both materials, the mass of both components was computed using the linear regression coefficients observed in the previous experiment via the calibration on pure material samples. Finally, the approximated mass of each material in each image was compared to the weighed masses.

### 2.5. Transformation from the FOV to Provide Setup-Independent
Measurements

The linear regression coefficients for the correlation between the segmented area of one material and its mass were dependent on the setup, as the area in the image increased when the working distance $L_{W}$ decreased.

Therefore, we also validated our method by calculating an equivalent spatial area of all particles and analyzed the correlation of that equivalent area to the particles mass. Any object size measured in the images can be converted to dimensions in the FOV using the relationship between focal length ($L_{F}$) and working distance ($L_{W}$). Given the size of a single pixel ($s_{P}$) on the sensor, a lateral length in the FOV can be calculated by:(4)$$d_{FOV}=\frac{s_{P}\;\cdot \;n_{P}}{L_{F}}\;\cdot \;L_{W}\;.$$
where $n_{P}$ denotes the number of pixels over which an object extends. In our experiments, we implemented the transformation from the area measured in number of pixels to the spatial area (in $mm^{2}$) by computing the radius of a circle with an area that is equivalent to the area covered by the segmented foreground pixels. This radius can be converted to millimeters, which leads to the absolute area of a circle with the equivalent radius in $mm^{2}$.

$L_{W}$ in our setup is (88.5±0.5)mm, the focal length of the camera $L_{F}$ according to the data sheet is 3.04mm, and the physical size of an individual pixel on the sensor $s_{P}$ is 1.12μm in both dimensions. $L_{F}$ and $s_{P}$ are taken from the camera module’s data sheet, which does not provide an uncertainty, but we assumed they were one to two orders of magnitude smaller than the uncertainty of $L_{W}$.

## 3. Results and Discussion

### 3.1. Evaluating Mass from 2D Images for Pure Material Samples

To evaluate whether our proposed method is suitable to estimate the actual mass of electrode particles from LIB in the recycling process, different experiments were performed. Our very first approach, which is not described in this paper, was imaging individual particles and correlating the segmented foreground area to their mass. This led to an insufficient correlation (r=0.79), likely due to the segmentation error, which was large in relation to the area of individual particles.

In our first relevant experiment, copper and aluminum particles were tested individually but with a larger number of particles in each sample. We found that the mass and projected area in 2D images clearly correlated even if the particles were not regularly shaped. The results for different samples are depicted in Figure 4. Here, we show the actual mass of the material samples determined by weighing versus the formal number of foreground pixels in relation to the overall number of pixels in our images. The diagram clearly indicates the correlation between the two variables. We prepared 10 pure samples of copper particles and 14 samples of aluminum particles. The linear regression analysis shows a steeper slope for the copper samples compared to the aluminum samples (Table 1). This was expected as the density of copper (8.92 g/cm${}^{3}$) is roughly 3.3 times higher than the density of aluminum (2.7 g/cm${}^{3}$). Since the offset of the intercept is relatively small compared to the slope, this method seems to be well extrapolatable. To receive the most accurate results, this method should be calibrated in a sensible working range.

Small deviations can occur due to the complex shape of the particles. The compaction of the particles results in a different roughness of the surface of the two materials. Due to different orientations of surface facets, light reflections can be different for each particle and even for different surface sections of one particle. This can lead to different identification of the particle edges. Light reflections on the background can also falsely count in the relative proportion of foreground pixels. The harsh pretreatment process, which compacts and folds the foils into compact particles, generates a certain distribution of the internal porosity. This also leads to some fluctuations in the relation of particle size and mass. However, as long as the pretreatment of the calibrated material and, finally, the measured material is identical, this error is assumed to be minor.

### 3.2. Detecting and Analyzing Mixed-Material Samples

With our second experiment, we tested the accuracy of our method when applied to real samples of unseparated mixed materials. Applying the color thresholding to samples of mixed materials can lead to unexpected results as the surface of the particles is not homogeneous in color, which can lead to wrongly segmented masks. As described in the previous chapter, we compared different methods to find the most accurate separation of materials. As seen in Table 2, there were significant differences in how well the approximated masses correlated with the real mass. The clustering approach did not achieve a sufficient correlation. The thresholding method worked well with a correlation coefficient of 0.984 for aluminum and 0.997 for copper. Since the mass of the overall sample was easy to determine (even inline), a third approach was to only compute the mass of one material component from the image analysis, weigh the entire sample, and subtract the computed mass to obtain the mass of the second material component (cumulative mass). This method worked slightly better for the copper component, so we continued using this approach for the copper particles.

In Figure 5, we display the true mass of both materials from our four mixed samples versus the approximated mass with the formerly described respective best method. For a perfect approximation, all data points should be on the diagonal line. For our samples, they were distributed around the line with relatively small uncertainties (compare Table 3). Table 3 also shows that when averaging the results of all samples, we achieved relative uncertainties lower than 2% for both materials. This is not only better than the human performance we usually observe (see Section 3.4) but also requires only a small fraction of the time after setting up the method.

### 3.3. Setup-Independent Results

As described in Section 2.5, the previously computed regression coefficients (Table 1) were bound to the exact setup we used. To make this method independent of the setup or specific parameters (e.g., the working distance $L_{W}$ of the imaging system), we transferred the area of particles measured in pixels into spatial areas using Equation (Equation 4) to compute a radius (in mm), which was equivalent to the radius (in pixel) of a circle with the same area as all foreground pixels. This approach increased the error since additional measurements with uncertainty were introduced (especially the working distance $L_{W}$). We found the correlation of the area in the field of view to be 0.952 for aluminum and 0.977 for copper with pure material samples, which was not significantly lower than the correlation using setup-dependent particle measurements in pixels. Those results suggest that the method can also be used to compare measurements from different setups as long as it is possible to average over multiple samples.

### 3.4. Comparison to the Existing Method

According to our experience, a skilled person is able to determine the mass composition with an accuracy of around 94% in roughly 65 h of work for 500 g of mixed particles. Adapting the shown method requires hardware costs of less than EUR 150 for a Raspberry Pi including the camera module. The hardware costs amortize after the first sample analysis. The human performance is exceeded by the system with an accuracy of >98%. This shows that a cost-effective, reliable, and quick method has been identified to analyze battery materials after recycling. The method offers a faster way to evaluate and adjust the separation process.

## 4. Conclusions

The evaluation of separation processes in recycling, in particular, the mass determination of the separated material streams is currently performed through manual sample separation. We introduced a method that enables the fast and accurate approximation of the mass composition of samples from aluminum and copper particles as they are found in the recycling process of lithium-ion batteries. The method can be automated after initial calibration and, therefore, is an important contribution towards making recycling processes and the corresponding performance evaluation more efficient. We showed that it is possible to segment the processed material with simple thresholding methods, providing sufficiently accurate information about the projected area of the particles present in an image. Furthermore, we showed that the segmented area, even though being impacted by both false positive and false negative segmented areas, can clearly be correlated to the mass of the particles, which enabled us to quantify the previous separation process with a higher accuracy than through manual sorting.

Since the material samples in the recycling process can differ from the ones we have shown, we also tested our segmentation method with different thresholds on the material from a battery cell with a different cathode material (Li-NCA instead of Li-NMC) and a different thickness of the individual layers, resulting in less reflective aluminum particles and a lower internal porosity after the compaction process. The correlation of the segmented areas to the material mass is comparable to the results shown but the differing internal porosity resulted in different regression parameters. Therefore, as expected, the method has to be recalibrated but is applicable on different materials as long as the initial color segmentation is sufficiently accurate. The introduced method enables users to save time and manual work after the initial set up, thus cutting costs as the system quickly amortizes and even exceeds the accuracy of manual sorting.

Both the illumination and the used background material could be optimized to improve the contrast between the particles and between the particles and the background. Additionally, the amount of false positive and false negative detected pixels could be optimized by more advanced postprocessing steps and by fine-tuning the morphological operations applied. Another important possible source of errors impacting the transformation from measurements in the images to spatial measurements is radial lens distortion. For an industrial use case with high accuracy requirements, the lens distortion could be determined and compensated mathematically. We plan to further investigate the effects of such improvements on our method in the future.

With recent advances in the deep learning-based segmentation of arbitrary objects (e.g., by Kirillov et al. [32] and Ke et al. [33]), these approaches could also be considered as an alternative way of finding the projected area of particles in the images. As stated previously, the state-of-the-art models we tested are not able to segment large quantities of objects within a single image. We are also considering this specific instance segmentation problem for further research, but we are focusing on the simpler approach of threshold segmentation, which seems more suitable for production implementation.

## Figures and Tables

**Figure 1 jimaging-09-00135-f001:**
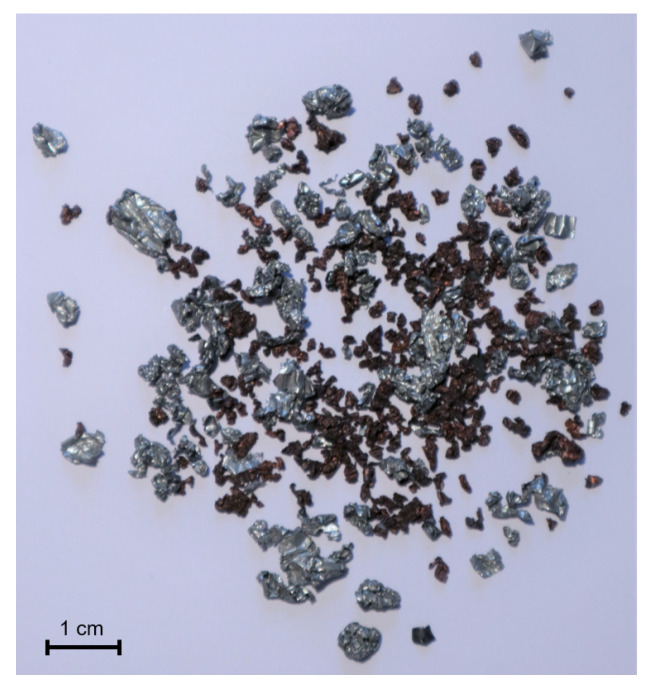
Aluminum and copper particles from a battery cell after processing.

**Figure 2 jimaging-09-00135-f002:**
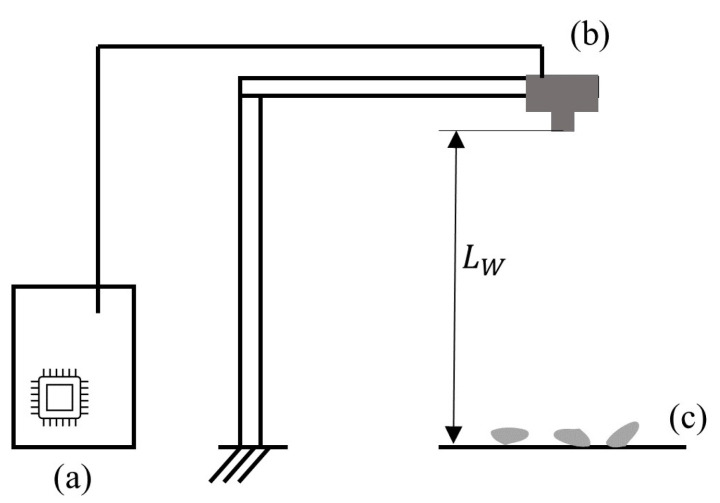
Schematic drawing of our experimental setup, displaying the Raspberry Pi (a), the camera module (b), and the particles placed on the background (c).

**Figure 3 jimaging-09-00135-f003:**
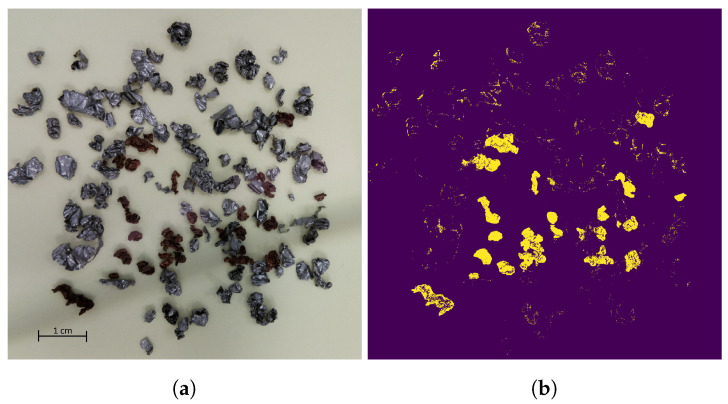
(**a**) Image of the mixed-material samples. (**b**) Resulting segmentation mask for the copper particles in the sample after color thresholding in the HSV color space. The segmentation mask clearly shows the copper particles. Due to irregular shapes and light reflections, the segmentation is superimposed with noise.

**Figure 4 jimaging-09-00135-f004:**
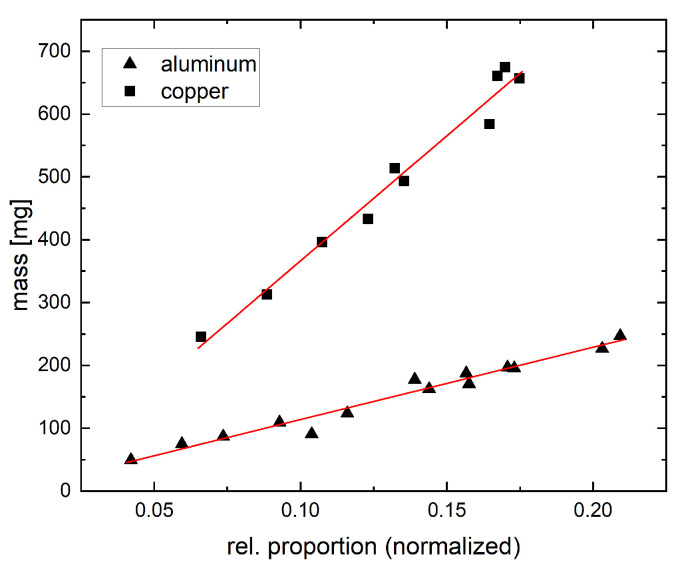
Correlation between the relative amount of foreground pixels in the segmented images and the mass of the respective particles.

**Figure 5 jimaging-09-00135-f005:**
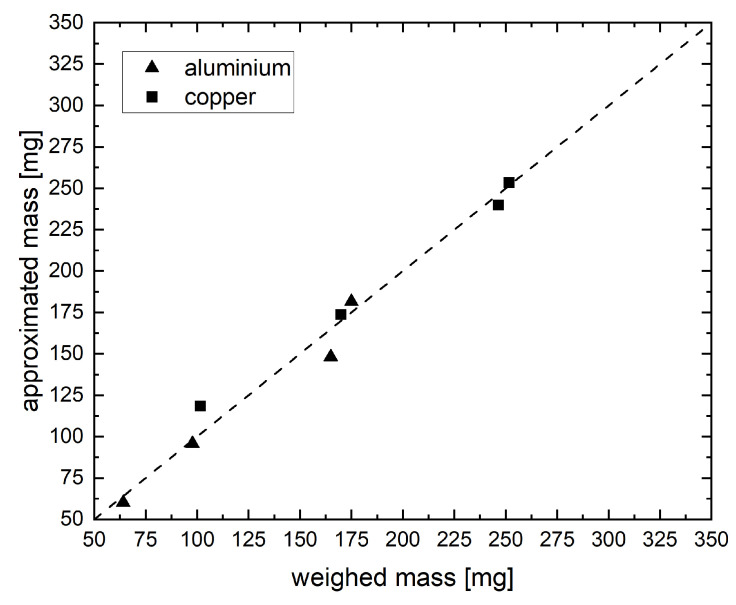
Weighed mass of both materials in the sample versus the approximated mass determined by weighing the cumulative mass and subtracting the mass of aluminum approximated using our method.

**Table 1 jimaging-09-00135-t001:** Linear regression analysis of the data shown in Figure 4.

Material	n	Slope	Intercept	$R^{2}$	Pearson’s r
Aluminum	14	1151.29	−1.4252	0.9684	0.98407
Copper	10	3980.25	−31.775	0.9772	0.98853

**Table 2 jimaging-09-00135-t002:** Pearson’s correlation *r* for each of the tested methods with the respective mass of both materials with the greatest correlation highlighted for each material.

Material	Clustering	Thresholding	Cumulative Mass
Aluminum	0.6512	**0.98431**	0.90318
Copper	0.8345	0.99718	**0.99781**

**Table 3 jimaging-09-00135-t003:** Weighed masses *m*, approximated masses $m^{\prime }$, and relative uncertainty δ of mixed samples of copper and aluminum from images determined using the segmentation of aluminum and subtraction from overall mass.

Sample	*m* ${}_{\mathrm{Al}}$	$m_{\mathrm{Al}}^{\prime }$	$\delta_{\mathrm{Al}}$	*m* ${}_{\mathrm{Cu}}$	$m_{\mathrm{Cu}}^{\prime }$	$\delta_{\mathrm{Cu}}$
(#)	(mg)	(mg)	(%)	(mg)	(mg)	(%)
1	165.0	150.16	8.99	101.6	116.44	14.6
2	64.1	61.96	3.35	169.8	171.94	1.26
3	97.8	97.65	0.15	251.6	251.75	0.059
4	175.0	183.19	4.68	246.4	238.2	3.33
Total	501.9	492.96	1.78	769.4	778.33	1.16

## Data Availability

The data presented in this study are available on request from the corresponding author.

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
