# Peer review of "Fast and Efficient Evaluation of the Mass Composition of Shredded Electrodes from Lithium-Ion Batteries Using 2D Imaging"

_2313-433X, 2023, doi:10.3390/jimaging9070135_

Round 1
Reviewer 1 Report
The manuscript describes a method based on image analysis for the quantification of the copper and aluminium particle content of recycled battery materials. This work is also a good example of how a direct, albeit relatively simple, approach can provide better success than an approach based on deep-learning techniques. The topic and the approach are thus both interesting but the manuscript requires some correction (see “Major issues”) before publication. Moreover, some (optional) improvements (see “Remarks and suggestion”) should improve the value and strength of the paper.
Major issues
- Abstract: The most interesting part of the manuscript (the image-based recognition and measurement of the material composition) is not addressed in the abstract. I strongly suggest reducing the contextualization part to a few lines and briefly describing the image analysis method and main results (especially with reference to human operator performance) to emphasize the value of the paper in the abstract.
- Experimental description. The image acquisition system needs to be better characterized to contextualize the results, and several missing parameters should be reported (e.g., camera spatial resolution, pixel bit depth, lens characteristics such as focal length and aperture, illumination system, background surface material, etc.). Another important parameter is the file format used for image registration (ideally .bmp or other uncompressed format) or an explicit statement that the image is analyzed directly from the raw capture (i.e., without any intermediate step of image compression/decompression).
- A figure illustrating the sample, the camera and the light source(s) relative positions with the relevant quotes should be added.
- It is not very clear to me the part related to the area measurement (rows 139-150) and the paragraph 3.3 (rows 206-214). Particularly, it is unclear the actual method used for the computation of projected area: it is calculated in square millimeters using the equation (4) as stated in the first part or it was calculated as segmented pixels as stated in the paragraph 3.3? These part should be better explained. See also the comment below relative to the camera calibration.
Remarks and suggestions
– The segmentation of copper and aluminium particles based on the chromatic difference is the most critical and interesting part. Here a possible major error source is the missing identification of particle-related pixels (i.e. false negative errors) as shown in figure 2b where several parts of the copper particles were erroneously attributed to the background. This is probably due to the very critical task of classifying the pixel as a background or particle and, at the same time, classifying particle-related pixels as copper or aluminum. I would then suggest evaluating the following alternative strategy based on the separation of the two tasks: first, separate all particles (both aluminum and copper) from the background as foreground pixels (e.g., using the brightness channel) and in a separate second step evaluate, for each particle found (i.e., for each connected cluster of foreground pixels) the classification as copper or aluminum (e.g., evaluating the color classification for all pixels in the particle cluster and assigning the entire particle to the most represented class).
Particle/background segmentation performed as a stand-alone process should give far better results. Moreover, simple morphological operations (such as hole filling) can improve the obtained segmentation. The background-particle segmentation can be further improved using a pale colored background and taking advantage of the chromatic information to reliably separe the background pixels from the foreground ones (the color of the background should be as different as possible from the colors of the copper and aluminum particles). This should improve the completeness of identification of particle-related pixels and thus the precision of the area evaluation.
- Instead of the (somewhat convoluted) procedure for area computation described in the manuscript (rows 117-121) consider to calculate the sample equivalent area of a single pixel and simply add, for each particle (or for the cumulative sample) the area of all attributed pixels. This approach has the interesting advantage that, in case of a significative geometric lens distorsion, a very simple compensation can be implemented estimating the area correction for each pixel on the basis of its position on the camera field (i.e. in case of the very common radial distorsion a correction function can be applied computing for each pixel the distance from the image optical center).
- In fig. 2 some shadowing is apparent. Consider, if possible, to implement a ring-shaped illuminator. If you want to stay in a low-budget implementation this can be realized with 10-12 inexpensive white LEDs mounted on a circular support around the lens (provided that the LED source is compatible with the color analysis process).
- If possible, it is suggested to verify the level of lens distorsion by imaging, in the same setup used for the study, a suitable target (ideally an optical calibration standard but also, at very minimum, a graph paper sheet). This is advisable because low cost lenses can suffer for significative radial (barrel/cushion) distorsion that can affect the area measurement in function of the position in the camera field. On the basis of the found distorsion it can be decided to compensate with some method (as before described) or to simply report the estimated error.
- It can be also better, if possible, to use a known dimensional reference in order to directly calculate the scale factor instead of rely on the nominal lens focal length and camera placement (the equation 4). This can be done in a separate calibration session or for every acquired image (including in the field a feature of known dimensions depicted on the background surface).
- Given that the aluminium/copper discrimination in based on color analysis consider also, if possible, to introduce a color reference method in order to guarantee constant and reproducible color values (expecially if in your setup there is a significant contribution of ambient light). A very cheap one is to use as sample background a white or grey ceramic tile with a matte surface in order to avoid parasitic reflections. This can provide for a very uniform and stable background and for a reference for color and exposure calibration (far better than a white paper sheet). If a colored background is used to enhance the particle/background discrimination, the use of a very unsaturated tint should allow the use of the same background as reference color.
- If I’ve correctly understood, a first experiment was attempted measuring individual particles but a significant error was found in the mass estimation from the projected area (rows 155-158). A second experiment was then performed evaluating the cumulative projected area from a sample composed by several particles, leading to better results (rows 159-171). This is very probably due to the averaging of the single particle errors, as suggested in the text. In this case the data from the individual particle measurement can be quite valuable and it can be interesting to quantify and comparing the found errors in individual measurement with the averaged error found in the collective measurement.
– The volume-area relationship for the particles should be (assuming the same form factor) a 3/2 power. It can be interesting to evaluate if this is the case for the data in Figure 3 and comment possible deviations.
Minor issues
- On my printer (Canon Imagerunner) the interpolation lines in Figures 3 and 4 appear very thin and barely legible. Consider specifying a greater thickness (e.g., 0.3 pt).
Reviewer 2 Report
1. Introduction should be more detailed. The goal of your introduction is to let your reader know the topic of the paper and what points will be made about the topic. Background needs to be more detailed, and what difficulties authors try to solve need to be more detailed. Authors only show one reference about the difficulty, more is needed to show the importance of this topic.
2. Add scale bar to Figure 2.
3. The total types of elements existing in the samples are few, authors should consider add extra experiments including more elements to show the the generalization of the proposed approach.
4. All calculations in the paper should include errors, no just model accruracy.
5. The conclusion paragraph should restate your thesis, summarize the key supporting ideas you discussed throughout the work, and offer your final impression on the central idea. This final summation should also contain the moral of your story or a revelation of a deeper truth.
6. Section 4 should be named Conclusion, if authors want they can have extra discussion section explaining and evaluating what you found, showing how it relates to your literature review and paper or dissertation topic, and making an argument in support of your overall conclusion.
Moderate editing of English language required.
Round 2
Reviewer 2 Report
Authors addressed most but not all comments. About the adding experiment comment there is no reply.
Minor editing of English language required
Author Response
We would like to thank the reviewer for his contribution to the scientific process. Regarding the third comment from the previous round of review ("The total types of elements existing in the samples are few, authors should consider add extra experiments including more elements to show the the generalization of the proposed approach."), we had previously commented that we believe there would not be additional benefit in adding more experiments with different elements.
The method is very dependent on optical differences of the segmented materials. Therefore, adding more experiments with different materials does not seem to be benefitial to us as the results would never be generalizable on any arbitrary combination of materials or elements. Instead we have mentioned in the "Conclusion" section, that we have performed similiar experiments with a different cathode material (Li-NCA instead of Li-NMC), which results in different regression results due to the lower internal porosity. Since the correlation of the segmented areas to the respective mass is comparable to the results shown in the manuscript, the method works on the Li-NCA cathode in combination with copper particles as well.